# Lung Metastasectomy: Where Do We Stand? Results from an Italian Multicentric Prospective Database

**DOI:** 10.3390/jcm13113106

**Published:** 2024-05-25

**Authors:** Marcello Carlo Ambrogi, Vittorio Aprile, Stefano Sanna, Sergio Nicola Forti Parri, Giovanna Rizzardi, Olivia Fanucchi, Leonardo Valentini, Alberto Italiani, Riccardo Morganti, Carlotta Francesca Cartia, James M. Hughes, Marco Lucchi, Andrea Droghetti

**Affiliations:** 1Department for Surgical, Medical, Molecular Pathology and Critical Care, University of Pisa, 56124 Pisa, Italy; 2Division of Thoracic Surgery, University Hospital of Pisa, 56124 Pisa, Italy; 3Multispecialistic Surgical Department, Private Forlì Hospitals, 47122 Forlì, Italy; 4Department of Thoracic Surgery, IRCCS University Hospital of Bologna, 40138 Bologna, Italy; 5Division of Thoracic Surgery, Humanitas Gavazzeni Hospital, 24125 Bergamo, Italy; 6Statistical Support Division for Clinical Studies, University Hospital of Pisa, 56124 Pisa, Italy; 7Division of Thoracic Surgery, Candiolo Cancer Institute, FPO-IRCCS, 10060 Candiolo, Italy

**Keywords:** lung metastasis, metastasectomy, lung surgery, thoracic surgery

## Abstract

**Background/Objectives**: The surgical resection of pulmonary metastases is considered a therapeutic option in selected cases. In light of this, we present the results from a national multicenter prospective registry of lung metastasectomy. **Methods**: This retrospective analysis involves data collected prospectively and consecutively in a national multicentric Italian database, including patients who underwent lung metastasectomy. The primary endpoints were the analysis of morbidity and overall survival (OS), with secondary endpoints focusing on the analysis of potential risk factors affecting both morbidity and OS. **Results**: A total 470 lung procedures were performed (4 pneumonectomies, 46 lobectomies/bilobectomies, 13 segmentectomies and 407 wedge resections) on 461 patients (258 men and 203 women, mean age of 63.1 years). The majority of patients had metastases from colorectal cancer (45.8%). In most cases (63.6%), patients had only one lung metastasis. A minimally invasive approach was chosen in 143 cases (30.4%). The mean operative time was 118 min, with no reported deaths. Morbidity most frequently consisted of prolonged air leaking and bleeding, but no re-intervention was required. Statistical analysis revealed that morbidity was significantly affected by operative time and pulmonary comorbidities, while OS was significantly affected by disease-free interval (DFI) > 24 months (*p* = 0.005), epithelial histology (*p* = 0.001) and colorectal histology (*p* = 0.004) during univariate analysis. No significant correlation was found between OS and age, gender, surgical approach, surgical extent, surgical device, the number of resected metastases, lesion diameter, the site of lesions and nodal involvement. Multivariate analysis of OS confirmed that only epithelial histology and DFI were risk-factors, with *p*-values of 0.041 and 0.031, respectively. **Conclusions**: Lung metastasectomy appears to be a safe procedure, with acceptable morbidity, even with a minimally invasive approach. However, it remains a local treatment of a systemic disease. Therefore, careful attention should be paid to selecting patients who could truly benefit from surgical intervention.

## 1. Introduction

The lungs constitute one of the most common sites of metastases, with an estimated 30% of patients with a variety of primary tumors developing pulmonary metastases. Since the first report published in 1965 about pulmonary metastasectomy by Thomford et al., numerous significant issues related to surgical treatment have been explored [1]. This includes analyses of indications, outcomes, different types of resections, and attempts to identify potential prognostic factors [2]. Today, lung metastasectomy is a routinely performed procedure and considered an integral component of a multidisciplinary treatment approach that can be tailored to individual cases. The foundational study by Pastorino et al. identified completeness of resection, disease-free interval, the number of metastases and primary tumor histology as major prognostic factors for long-term survival [3]. However, this landmark study dates back to 1997. In the ensuing decades, advancements in surgical techniques, including the application of minimally invasive approaches, as well as the development of more effective chemotherapy regimens, have expanded the role of surgery [4,5,6]. A recent study by Internullo et al., highlighted the increasing prevalence of lung metastasectomy procedures in European counties [7]. In this context, we present the findings of a prospective multicentric national registry, documenting the current practices within our healthcare system. The objective of this multicenter prospective registry was to gather high-quality data within a defined timeframe, focusing on optimal resection techniques, the type of surgical device and surgical approaches. The secondary endpoint was the analysis of potential risk factors influencing overall survival (OS).

## 2. Materials and Methods

This was a retrospective analysis of data collected prospectively and consecutively from the national multicentric registry organized in Italy. The registry included all patients who underwent lung metastasectomy with curative intent within a 5-year interval from 2010. The median follow-up period was 51 months, ranging from 3 to 105 months. The exclusive diagnostic purpose, particularly in patients with multiple lesions, was considered an exclusion criterion.

A detailed database was designed in order to provide each participating hospital with a simple tool for data recording. For all patients, the following variables were documented in the registry: age, gender, performance status (ECOG score: the Eastern Cooperative Oncology Group), FEV1 (Forced Expiratory Volume in 1 s: liters and percentage of predicted), FVC (Forced Vital Capacity: liters and percentage of predicted), the histology of the primary tumor, the time between the first intervention for primary cancer and lung metastasectomy (months), previous chemotherapy or radiotherapy; cN status (based on Computer Tomography (CT) scanning and Positron Emission Tomography (PET) scanning), the number and size of lung metastases (in the case of more than one metastasis, the diameter of the largest one was recorded), the site of lung metastases (unilateral or bilateral), the type of operation (pneumonectomy, lobectomy, segmentectomy, wedge resection), the type of surgical approach (VATS or thoracotomy), the type of surgical technique (stapler, energy device, standard electrocautery or laser), the operative time (minutes), eventual residual disease, post-operative stay, morbidity, 30-day mortality, pN status, overall survival (defined as the time from first lung metastasis surgery and last follow-up or death) and the sites of new metastases. Laser ablation was classified as a wedge resection, representing a non-anatomical approach.

### Statistical Analysis

Continuous variables were expressed as mean and standard deviation (SD) or median and range, while categorical variables were presented as frequency.

Regarding statistical analysis, patients who underwent planned sequential resections within 1 month were considered to have had a single metastasectomy, not redo surgery. When multiple resections were performed during the same procedure, the type of resection was classified based on the largest type. Operative time, morbidity and mortality were calculated based on overall surgical interventions, including those planned within 1 month. In this case, survival analysis was calculated from the date of the first metastasectomy.

For patients who underwent new lung resection for metastases developed during the follow-up period, operative time, morbidity and mortality were recorded but not included in the statistical analysis of morbidity. Survival analysis was conducted from the date of the first pulmonary metastasectomy.

Multivariate analysis, based on binary logistic regression with a step-wise method, was employed to explored possible relations between morbidity and age, gender, the presence of cardiovascular comorbidities (yes vs. no), the presence of pulmonary comorbidities (yes vs. no), ECOG score, adjuvant therapies (yes vs. no), the type of resection (wedge vs. anatomical resection: segmentectomy, lobectomy or pneumonectomy), the type of approach (VATS vs. thoracotomy), the type of device (mechanical stapler vs. laser, energy device or standard electrocautery), the application of laser (yes vs. no), lymphadenectomy (yes vs. no), operative time, side approach (unilateral vs. bilateral), the number of resected lesions (1 vs. >1), DFI (<24 vs. >24 months) and primary histology (epithelial vs. other types)

Additionally, we investigated potential relationships between residual disease (R0 vs. R1/R2) and a lesion’s dimension, the type of resection (wedge vs. anatomical resection: segmentectomy or lobectomy or pneumonectomy), the type of device (mechanical stapler vs. laser, energy devices vs. electrocautery), the application of laser (yes vs. no), the type of approach (VATS vs. thoracotomy) and the number of resected lesions (1 vs. >1). For the analyses, we applied the χ2 test or Mann–Whitney test, as appropriate, for continuous data and categorical measures, respectively.

Overall survival (OS) was estimated using the Kaplan–Meier method. Univariate analysis was conducted to explore possible correlations between OS and age, gender, DFI (<24 vs. >24 months), the type of resection (wedge vs. anatomical resection: segmentectomy, lobectomy or pneumonectomy), the type of device (mechanical stapler vs. laser/energy devices/electric scissor), the application of laser (yes vs. no), the type of approach (VATS vs. thoracotomy), the number of lung metastases resected (1 vs. >1), the site of the lesion (unilateral vs. bilateral), residual disease (R0 vs. R1/R2), the size of the larger lesion, the histology of primary cancer (epithelial vs. others; colon vs. others) and N status (cN0/pN0 vs. cN1/pN1/pN2). In addition, multivariate analysis, using Cox regression, was performed for factors with significance < 0.1 at univariate analysis. The hazard ratio and corresponding 95% confidence intervals (CIs) were reported for covariates. A *p*-value of <0.05 was considered statistically significant.

## 3. Results

A total of 461 patients were enrolled in the registry, comprising 258 (56%) males and 203 (44%) females, with a mean age of 63.1 years (SD 11.4). Key clinical features are summarized in Table 1. The primary cancer site was most frequently the colon (45.8% of cases), followed by kidney (10.4%) and sarcoma (7.6%). Among the patients, 223 (48.4%) underwent chemotherapy and/or radiotherapy after the resection of the primary cancer. The mean DFI was 47.3 months (SD 40.1). In the majority of cases (63.6%), patients presented with a single lung metastasis. Bilateral lung metastases were observed in 23 patients: 14 patients underwent bilateral resection during the same procedure, while 9 patients were scheduled for sequential lung resections on both sides at a 1-month interval. This resulted in a total of 470 surgical interventions (4 pneumonectomies, 46 lobectomies/bilobectomies, 13 segmentectomies, 407 wedge resections) and a total of 957 resected metastases (median 1, range 1–36). A minimally invasive approach was chosen in 143 cases (30.4%), with 142 undergoing Video-Assisted Thoracoscopic Surgery (VATS) and 1 utilizing a robot-assisted approach. Notably, the VATS approach was employed in 39.2% of patients with single lung metastasis (115/293).

The mean operative time was 118 min (range 15–435). Mechanical staplers were predominately used in most cases (69%), either alone or in conjunction with other energy devices. In 31% of cases, coagulative tools were used as sole device for parenchymal resection; in particular, a laser was applied in 106 cases (23% of patients). Lymph node sampling or lymphadenectomy was performed in 220 patients, with 175 evaluated as cN0 during the preoperative course, and was pN0 in 164 cases, pN1 in 7 cases and pN2 in 4 cases. Of the 45 patients considered cN1 who underwent lymph node sampling or lymphadenectomy, 37 were pN0 and 8 were inpN1. Table 2 summarizes post-operative complications that occurred in 83 (17.6%) cases out of 470 surgical interventions: the most frequent complications were air-leaking, with 26 cases, and bleeding, with 21 cases. Some patients experienced two or more complications, but none required surgical re-intervention; instead, they were successful managed with medical therapies. No deaths occurred.

The diameter of the larger lesion was between 1 and 2 cm in 47.9% of cases and less than 1 cm in 26.9% of cases (Table 2). Macroscopic complete resection was obtained in all cases except 13 (2.09%), of which 9 had macroscopic residual disease (R2) because a complete resection would have necessitated sacrificing lung parenchyma incompatible with the patient’s respiratory functional reserve. Microscopic residual disease (R1) was observed in four patients (0.87%) at pathological examination at the resection borders, with three cases occurring after stapler resection and one case after laser resection).

During the follow-up period, 22 (4.8%) patients underwent surgical resection of new lung metastases (3 lobectomies and 19 wedge resections). Eight patients were lost at follow-up and consequently excluded from the statistical analysis of OS. The median follow-up period was 51 months (range 3–105).

### Statistical Analysis

In the examination of post-operative complications, we identified a statistically significant correlation between operative time and morbidity (*p* = 0.008). Additionally, the presence of pulmonary comorbidities emerged as a significant factor affecting morbidity (*p* = 0.021). However, no correlations were observed between morbidity and other variables, including age, gender, ECOG score, cardiovascular disease, adjuvant therapy, the type of surgical intervention, the type of surgical approach, the type of surgical device, the side approach, the number of resected lesions and lymphadenectomy. Furthermore, no higher complication rate was observed for patients treated with the application of laser (*p* = 0.419) (Table 3).

We also explored potential correlations between the presence of residual disease and various factors, such as lesion diameter, the number of resected lesions, the type of surgical resection, the type of surgical approach, the type of device and the application of laser. However, no statistically significant correlation was observed (Table 4).

Turning to the analysis of possible risk factors affecting OS, univariate analyses (Table 5) revealed no correlation between OS and variables such as age, gender, the type of surgical resection, the type of approach, the type of surgical device, the number of treated lesions, the site of lesions (unilateral vs. bilateral), lymphadenectomy/lymph node sampling (yes vs. no), residual disease (R0 vs. R1/R2), lesion diameter and N status (cN0/pN0 vs. cN1/pN1/pN2). However, a significant difference for better OS was noted in patients with a DFI greater than 24 months (*p* = 0.005): the 1-, 3- and 5-year OS rates were 96%, 75% and 58% for DFIs less than 24 months and 99%, 80% and 70% for DFIs > 24 months, respectively. Furthermore, epithelial histology significantly influenced OS (*p* < 0.001). The 1-, 3- and 5-year OS rates were 97%, 78% and 67% for epithelial metastases and 87%, 60% and 44% for other histologies (sarcoma, germ cell, melanoma), respectively. In univariate analysis, colon primary histology also demonstrated a significantly better OS compared to all other histologies (sarcoma, germ cell, melanoma) (Table 5).

In multivariate analysis, only epithelial histology and DFI continued to be significant factors influencing OS, with *p*-values of 0.041 (Figure 1) and 0.031 (Figure 2), respectively (Table 5).

## 4. Discussion

The lung is one of the most frequent sites for tumor spread. The first papers on the surgical resection of lung metastases date back to the early 20th century [8,9,10]. To date, the resection of lung metastases has become an important part of the daily routines of thoracic surgeons, as evidenced by a study of 2008, investigating the clinical practices of members of European Society of Thoracic Surgeons (ESTS) [7]. Internullo et al. emphasized the role of lung metastases surgery within the framework of personalized treatment for advanced cancer. However, the criteria for surgical resection remain rooted in the context of 1997, when Pastorino et al. reported the results of the International Registry of Lung Metastases: surgical resection should be reserved for selected patients with complete control of primary tumor and no extra-thoracic disease [3]. However, precise guidelines are still lacking and evidence about the survival of patients with features making them eligible for, but who did not actually undergo, metastasectomy are missing. The Pulmonary Metastasectomy in Colorectal Cancer (PulMiCC) trial was the only randomized controlled trial that tried to answer this question for metastases from colorectal cancer. However, this study was early closed due to poor and worsening recruitment and the small number of participants, which precluded a conclusive answer to the research question [11].

Here, we present the outcomes of a prospective multicentric Italian database, contributing valuable insights into the current practices within our healthcare system. In our study, wedge resection emerged as the predominant procedure (86.6%) for resecting pulmonary metastases, according to the data of the literature [7,12]. As a baseline, the intent of surgery was to preserve lung parenchyma, allowing subsequent resections for new lung metastases. In fact, as occurred in the studies of Casiraghi et al. [13] and Kandioler et al. [14], as well as in our series, 22 patients underwent surgical resection of new lung metastases. Anatomical resections, including four pneumonectomies, were reserved for larger or central lesions, with the aim being to achieve a complete resection, as supported by many papers in the literature [12,15]. Nevertheless, in our study, the type of surgical resection did not impact post-operative morbidity (Table 3). This evidence can be likely attributed to the high prevalence of wedge resections with respect to anatomical resections and meticulous preoperative patient selection.

Post-operative outcomes were favorable, with an overall morbidity of 17.6% and a 30-day mortality rate of 0%, consistent with the literature [13,16]. Multivariate analysis linked surgical morbidity to clinical factors (the presence of pulmonary disease) and a procedural aspect (operative time), as reported in Table 3 and by Suksompong et al. [17]. Despite these observations, a universally applied and validated risk predictor model for thoracic surgery, akin to other specialties, is still lacking [18].

Wedge resections in our series were primarily performed with stapler, either as the sole device or in addition to coagulating tools. It is considered that the use of devices other than staplers may be advantageous for multiple lung resections (up to 36 in our series) and preserving lung function, aligning with previous studies by Rolle et al. [4] and Kodama et al. [19]. In our series, laser was applied in 31% of cases, with no significant difference in terms of morbidity (*p* = 0.419, Table 3) with respect to other devices. In addition, no difference in terms of residual disease was observed in the case of laser application compared to other devices (*p* = 0.182, Table 4).

Regarding the surgical approach, VATS was applied in 30.4% of procedures, with no impact on post-operative morbidity. This percentage aligns with the data of the literature, especially if we considered surgical procedures that were performed between 2011 and 2015 [3]. However, the debate over the best surgical approach for lung metastases continues, with some studies recommending manual palpation of the lung, while others emphasize the benefits of minimally invasive approaches such as VATS [13]. In 1996, McCormack et al. published a study on lung metastases, recommending manual palpation of the lung [20]. Nevertheless, during the few last decades, many improvements in imaging technique were developed (i.e., high-resolution CT), and several papers have been published describing the successful use of preoperative labeling techniques (with hookwire, radionuclide or microcoil positioning) for lung nodule identification during thoracoscopy [21,22,23] in order to avoid the necessity of open thoracotomy. Therefore, the role of VATS in the resection of lung metastases has been expanded. Even if open thoracotomy provides a better field of palpation with an advantage in lung parenchyma sparing, a minimally invasive approach is associated with many clinical benefits [5,16,24,25].

Recent reviews, including one by Rusidanmu et al., considered VATS a good surgical technique for treating resectable oligo-metastatic lesions in the lungs [25]. In the present study, almost 40% of patients with a single lung lesion were treated with VATS. More recently, Claramunt et al. observed no significant difference in ipsilateral recurrence rates between VATS and open surgery in the treatment of colorectal cancer lung metastases. They underlined that the VATS approach was acceptable whenever complete resection can be ensured, and conversion to open surgery was indicated when lesions identified preoperatively were not found or when technical problems encountered may compromise surgical margins [26].

Comparative studies, such as the one conducted by Carballo et al. in 2009, demonstrated the safety and efficacy of both open and VATS approaches for lung metastasectomy, where both procedures appeared safe and efficacious, with no increased number of recurrent thoracic lesions developed during follow-up for patients treated with VATS: the non-inferiority analysis of 5-year overall survival demonstrated that VATS was equivalent to thoracotomy [6]. The current study also found no significant difference in OS between patients treated with VATS and open thoracotomy (*p* = 0.239). A recent review by Meng et al. considered VATS to be an alternative surgical approach for lung metastasectomy; however, it is acknowledged that further prospective studies are needed to identify the indications for VATS in patients with pulmonary metastases [27].

With regard to the type of surgical resection and type of surgical device, no significant correlation with OS was observed (Table 5), consistent with the existing literature [13]. Probably, this is related to the appropriate surgical procedure that was chosen, case by case, based on the anatomic location and extent of the disease in order to achieve a complete resection [3,28]. Cases involving lobectomies or pneumonectomies were infrequent and reserved for situations in which wedge resection was not feasible, such as lesions located deep near the hilum. Additionally, laser technology facilitated the resection of numerous metastases (up to 36 in our series) while preserving lung parenchyma [4].

Contrary to some studies emphasizing the independent prognostic significance of lesion size, our findings, along with others, did not establish a significant relationship between survival and the size of larger lesion [29,30,31,32,33]. It is suggested that the presence of residual disease rather than the size of the lesion may influence OS, as underlined in the corner-stone studies of Pastorino et al. [3] and Rush et al. [28]. In the present study, this evidence did not reach statistical significance (*p*-value = 0.099), probably due to the small number of R1/R2 disease cases (2.9%).

In addition to the diameter of the lesion, the number of metastases did not impact on OS in this series, consistent with some studies and in contrast to others. In the literature, some authors reported that the multiplicity of lung metastases was a poor prognostic factor [34,35]. Pfannschmidt et al. and Meacci et al. demonstrated that patients with solitary metastases from colon cancer and renal cell cancer, respectively, had significantly better survival than those with multiple ones [31,36]. On the contrary, other authors did not find a significant relationship between prognosis and the number of metastases akin to that in our series [6,13]. Inoue et al. found no significant difference in survival between patients with solitary and multiple lesions, suggesting that occult micrometastases might have existed at the time of metastasectomy in patients considered to have a solitary lesion [37]. Therefore, the presence of more than one metastatic lesion cannot be considered a contraindication, as recently reported in the paper of Internullo et al., where 85% of ESTS members did not consider multiple lesions a contraindication [7].

Similar to the number of nodules, contradictory data exist regarding the distribution of metastases (unilateral vs. bilateral) as a prognostic factor affecting OS. Inoue et al. reported that unilateral location was an independent predictor of longer survival, and Chen et al. reported no long-term survivors in patients with bilateral lesions [37,38]. On the other hand, McCormack et al. showed no significant difference in survival with regard to the distribution of lung nodules, suggesting that patients with bilateral lesions may benefit from metastasectomy, as well as those patients with ipsilateral multiple lesions [20]. Carballo et al. reported no significant difference in OS for patients undergoing bilateral compared with unilateral nodule resections (*p*-value = 0.40) [6]. The same result was also observed by Rolle et al. in the analysis of the complete resection of lung metastases in a heterogeneous series [4]. In this context, as we also observed, bilateral lesions do not seem to be a contraindication for lung metastasectomy if they can be completely resected [7].

The issue of lymph node dissection remains hotly debated, with evidence suggesting that patients with lymph node metastasis exhibit a poor prognosis, regardless of the type of malignancy involved [3,28,39]. In a recent meta-analysis on lung metastases from colorectal cancer performed by Gonzalez et al., hilar and/or mediastinal lymph node involvement was a prognostic factor of poor outcomes [40]. In this study, a tendency toward worse survival was noted for patients with lymph node involvement (*p* = 0.075) (cN1, pN1, pN2 status) with respect to those with negative lymph node (cN0/pN0). However, the small sample size of patients with positive lymph nodes and the variability in lymph node sampling/dissection practices may have influenced the results. In a survey among ESTS members, lymph node sampling was routinely performed by 55.5% of responding surgeons [7]. With regard to systematic nodal dissection with therapeutic intent, even if some authors observed long-term survivors after lymphadenectomy despite their metastatic nodal involvement, no evidence of impact on survival exists [41]. We also observed no advantages in terms of OS in patients who underwent lymphadenectomy. Among ESTS members, only 13% of surgeons routinely performed complete mediastinal lymphadenectomy, while 32.2% performed neither lymph node sampling nor dissection [7].

DFI emerged as a significant factor affecting survival in both univariate and multivariate analysis (Table 5). Patients with a DFI > 24 months demonstrated better survival compared to those with a DFI < 24 months. The 1-, 3- and 5-year OS rates were 99%, 80% and 70% for patients with a DFI > 24 months and 96%, 75% and 58% for patients with a DFI < 24 months, respectively (Figure 2). This finding is consistent with the literature, where a prolonged DFI is considered an independent prognostic factor for identifying patients who may benefit from pulmonary metastasectomy [3]. As in our series, Onaitis et al. and Gonzalez et al. demonstrated lung metastases from colorectal cancer, and a prolonged DFI was associated with a favorable treatment outcome [34,40]. Also, Meacci et al. reported a significantly poorer survival for synchronous metastasis from renal cell carcinoma [31]. However, not all investigators showed that a short DFI correlated with a poor prognosis after metastasectomy [15,33]. In their systematic review, Pfannschmidt et al. did not find DFI to be a significant prognostic factor [36]. Although various cut-off values for defining the short DFI might have affected the analysis of these studies, considering the clinical behavior of tumors, a short DFI represents an early dissemination of metastatic disease, which implies more aggressive tumor biology and worse overall survival. Thus, it may be reasonable that a short DFI might be a poor prognostic factor, but it is not an absolute contraindication, as believed by the majority of the surgeons in the paper by Internullo et al. [7].

This study also explored the impact of primary tumor origin on OS. Better OS was observed for epithelial cancer with respect to other histologies, both for univariate and multivariate analysis (Table 5). In addition, the subgroup analysis of colorectal cancer and non-colorectal cancer showed a significant difference in OS with univariate methods (*p* = 0.004), but this significance was not maintained in multivariate analysis (*p* = 0.091). This is consistent with the literature, where the primary tumor origin has been shown to influence survival, with better survival rates in epithelial cancers than in sarcomas or melanomas [3]. Hirai et al. showed that colorectal cancer patients had a better survival rate than patients with other primary organs involved (*p* = 0.003) [42]. However, even if it is well known that melanoma and sarcoma are associated with poor prognosis, tumor histology does not represent a contraindication to lung metastasectomy in most cases, as reported in the ESTS survey [5,7].

## 5. Conclusions

In conclusion, this multicenter Italian study suggests that lung metastasectomy is a safe procedure with acceptable morbidity. Some patients who undergo pulmonary metastasectomy survive for long periods, and some are completely cured, even when the tumors spread hematogenously to the lungs. However, this study emphasizes the need for paying careful attention to the surgical treatments for patients with stage IV disease, and three issues must be emphasized with regard to patient selection.

Firstly, data on lung metastasectomy are often supported by studies on mixed cancer types, adding difficulty to deducing precise guidelines due to the heterogeneity of the biological behavior of different primary tumors. Secondly, case selection is based on known favorable prognostic indicators [3], potentially creating a bias for interpreting results [13,20]. The only randomized trial on lung metastases from colorectal cancer (PulMiCC) was stopped early because of recruitment difficulties; thus, it was unable to clarify the value of pulmonary metastasectomy [7]. Therefore, only observational studies guide thoracic surgeons in their clinical decision-making for pulmonary metastasectomy for a single patient. Lastly, this study points out that only physicians who believe in surgical treatment will refer patients to surgeons, creating potential bias in the data toward patients operated on by surgeons who believe in local treatments for a systemic disease.

## Figures and Tables

**Figure 1 jcm-13-03106-f001:**
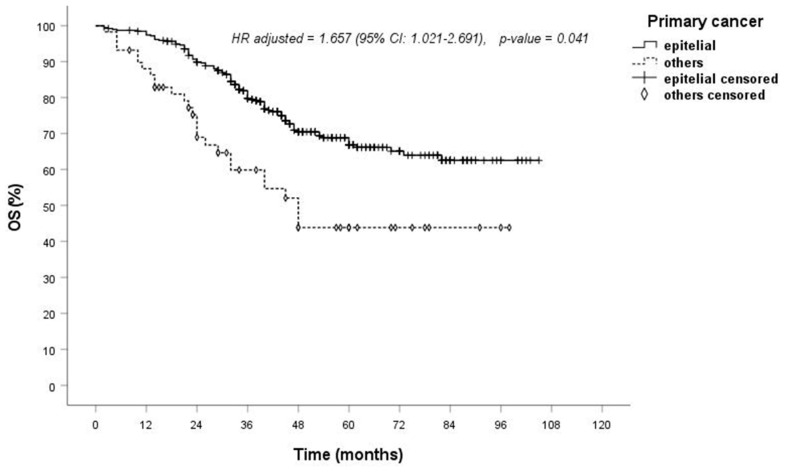
Overall survival according to the histology of primary cancer.

**Figure 2 jcm-13-03106-f002:**
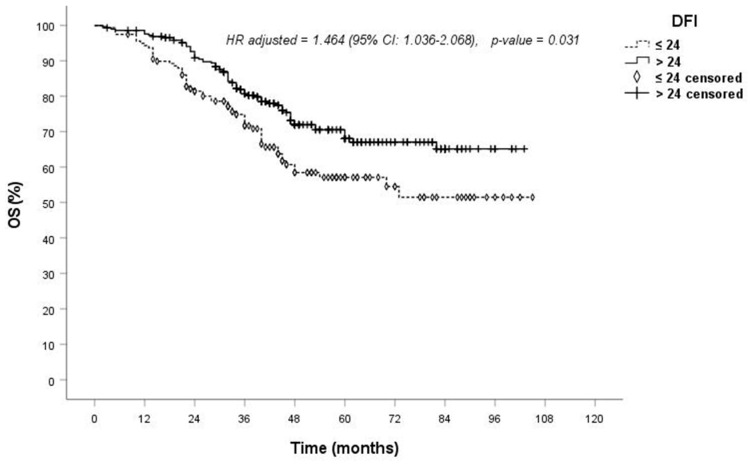
Overall survival according to the disease-free interval (DFI) from primary tumor treatment to metastasectomy.

**Table 1 jcm-13-03106-t001:** Clinical features of enrolled patients.

Variables	Number (%)
Age, years mean (±SD)	63.1 (±11.4)
Gender	
M (%)	258 (56%)
F (%)	203 (44%)
ECOG § n (%)	
0	403 (87.4%)
1	42 (9.1%)
2	16 (3.5%)
3	0
Cardiovascular comorbidities °	
n (%)	228 (49.5%)
Pulmonary comorbidities °°	
n (%)	54 (11.7%)
FEV1 % of predicted	
mean (SD)	101.1% (19.5)
FVC % of predicted	
mean (SD)	102.1% (18.8)
DFI (months)	
mean (SD)	47.3 (40.1)
DFI > 24 months n (%)	299 (64.9%)
DFI ≤ 24 months n (%)	162 (35.5%)
Adjuvant therapy *	
yes	223 (48.4%)
no	238 (51.6%)
Histology n (%)	
epithelial	401 (87.0)
sarcoma	35 (7.6%)
melanoma	21 (4.6%)
germinal cell	4 (0.8%)
Histology n (%)	
colon	211 (45.8%)
non-colon	250 (54.2%)
Clinical N status **	
cN0	381 (82.6%)
cN1	80 (17.4%)

§ ECOG (The Eastern Cooperative Oncology Group) performance status scale; ° cardiovascular comorbidities defined as coronary artery disease, myocardial disfunction, or prior coronary bypass/stenting and/or previous surgical intervention/stent positioning for vascular disease; °° pulmonary comorbidities defined as the presence of chronic obstructive pulmonary disease (COPD) or bronchiectasis or pulmonary fibrosis; FEV1% = Forced Expiratory Volume in 1 s; FVC = Forced Vital Capacity; DFI = disease-free interval defined as time between first intervention for primary cancer and lung metastasectomy; * adjuvant therapy defined as previous chemotherapy, immunotherapy or radiotherapy; ** clinical status based on CT and PET scanning.

**Table 2 jcm-13-03106-t002:** Operative and post-operative features.

Variables	Number (%)
Pathological dimension of larger lesion n (%)	
<1 cm	124 (26.9%)
1– <2 cm	221 (47.9%)
2– <3 cm	73 (15.8%)
3– <4 cm	17 (3.7%)
≥4 cm	26 (5.6%)
Number of lesions n (%)	
1 lesion	293 (63.6%)
more than 1 lesion	168 (36.4%)
Side of the lesions n (%)	
unilateral	438 (95.0%)
bilateral	23 (5%)
Type of approach * n (%)	
VATS	143 (30.4%)
open	327 (69.6%)
Type of surgical procedure * n (%)	
wedge resection	407/470 (86.6%)
segmentectomy	13/470 (2.8%)
lobectomy/bilobectomy	46/470 (9.8%)
pneumonectomy	4/470 (0.8%)
Operative time (min) * Mean (SD)	118 (64.3)
Surgical device n (%)	
mechanical stapler	321 (69.6%)
others	140 (30.4%)
Lymph node sampling/dissection n (%)	
yes	220 (47.7%)
no	241 (52.3%)
Pathological N status	
pN0	201/220
pN1	15/220
pN2	4/220
Post-operative complications * ° n	
air leaking	26
bleeding	21
pleural effusion	19
supraventricular arrhythmia	15
pulmonary embolism	1
others minor	40
Post-operative stay (days) * mean (SD)	4.4 (2.7)
Radicality n (%)	
R0	448 (97.1%)
R1	4 (0.9%)
R2	9 (2.0%)

VATS = Video-Assisted Thoracic Surgery; lymphadenectomy defined as lymph node dissection or sampling; * = calculated on a total of 470 surgical interventions; ° = some patients had more than one post-operative complication.

**Table 3 jcm-13-03106-t003:** Multivariate analysis of predictive factors affecting morbidity, based on binary logistic regression performed with a stepwise method.

Factor	RC	OR	OR (95% CI)	*p*-Value
Operative time ^a^	0.004	1.004	1.001–1.008	0.021
Pulmonary comorbidities ^b^	0.881	2.411	1.253–4.639	0.008
Gender ^b^				0.615
Age ^a^				0.353
Cardiovascular comorbidities ^b^				0.850
FEV1% of predicted ^a^				0.103
ECOG ^a^				0.232
Surgical procedure ^b^				0.479
Surgical approach ^b^				0.656
Surgical technique ^b^				0.897
Laser ^b^				0.419
Side ^b^				0.599
Lymphadenectomy ^b^				0.280
Number of lesions ^b^				0.119
DFI 24 ^b^				0.555
Primary histology ^b^				0.327
Adjuvant therapies ^b^				0.563

^a^ Continuous variable. ^b^ Categorical variables. Cardiovascular comorbidities defined as coronary artery disease, myocardial disfunction, or prior coronary bypass/stenting and/or previous surgical intervention/stent positioning for vascular disease (yes vs. no); pulmonary comorbidities defined as the presence of chronic obstructive pulmonary disease (COPD) or bronchiectasis or pulmonary fibrosis (yes vs. no); FEV1% = Forced Expiratory Volume in 1 s; ECOG performance status (The Eastern Cooperative Oncology Group); surgical procedure (wedge resection vs. anatomical resection: segmentectomy, lobectomy, pneumonectomy); surgical approach (VATS vs. open thoracotomy); surgical technique (mechanical staplers with or without other coagulating devices vs. coagulating tools as sole device); laser application (yes vs. no); lymphadenectomy: lymph node dissection or sampling (yes vs. no); the number of resected metastases (1 vs. >1); primary histology (epithelial vs. other histologies); DFI 24 (≤24 vs. >24); adjuvant therapies (yes vs. no).

**Table 4 jcm-13-03106-t004:** Analysis of factors affecting the presence of residual disease calculated with χ^2^ and Mann–Whitney tests *, where appropriate.

Factor	*p*-Value
Diameter *	0.289
Surgical procedure 407 wedge resections vs. 63 anatomical resections	0.535
Surgical approach 143 VATS vs. 327 open approaches	0.536
Surgical technique 324 Stapler ± coagulating devices vs. 146 solo coagulating devices	0.071
Laser application 106 laser applications vs. 364 cases of no laser use	0.182
Number of lesions ^#^ 293 patients underwent surgery for a single lesion vs. 168 patients operated on for multiple metastases	0.889

* ccontinuous variable; Surgical procedure (wedge resection vs. anatomical resection: segmentectomy, lobectomy, pneumonectomy); surgical approach (VATS vs. open thoracotomy); surgical technique (mechanical staplers with or without other coagulating devices vs. coagulating tools as sole device); laser application (yes vs. no); the number of resected metastases (1 vs. >1). The VATS group also included one robotic procedure. ^#^ calculated on a total of 461 patients.

**Table 5 jcm-13-03106-t005:** Univariate and multivariate analysis of factors affecting OS; all factors with a significance < 0.1 at univariate analysis were processed with the multivariate method.

Factor	HR (95%CI)	*p*-Value	RC	HR (95%CI)	*p*-Value
Gender ^b^	0.702 (0.495–0.995)	0.047	−0.310	0.733 (0.514–1.046)	0.087
Age ^a^	1.005 (0.990–1.021)	0.512			
DFI 24 ^b^	1.629 (1.160–2.287)	0.005	0.381	1.464 (1.036–2.068)	0.031
Side ^b^	0.867 (0.439–0.860)	0.733			
Lymphadenectomy ^b^	1.078 (0.771–1.508	0.659			
Surgical technique ^b^	0.991 (0.687–1.429)	0.960			
Primary histology ^b^	2.273 (1.490–3.465)	<0.001	0.505	1.657 (1.021–2.691)	0.041
Colon histology ^b^	0.614 (0.439–0.860)	0.004	−0.332	0.717 (0.488–1.054)	0.091
Surgical procedure ^b^	1.234 (0.775–1.965)	0.376			
Laser application ^b^	0.819 (0.537–1.248)	0.353			
Pathological dimension ^b^	1.144 (0.978–1.338)	0.092	0.079	1.082 (0.924–1.268)	0.327
Residual disease ^b^	1.883 (0.830–4.270)	0.130			
Surgical approach ^b^	0.941 (0.657–1.347)	0.740			
Number of resected lesions ^b^	1.228 (0.870–1.735)	0.242			
N status ^b^	1.482 (0.961–2.287)	0.075	0.192	1.211 (0.770–1.906)	0.407

^a^ Continuous variable. ^b^ Categorical variables; gender: male = 0, female = 1; DFI > 24 = 0, DFI ≤ 24 = 1; side: unilateral = 0, bilateral = 1; lymphadenectomy (lymph node sampling/dissection): no = 0, yes = 1; surgical technique: laser, energy device or electrocautery = 0, mechanical stapler = 1; primary histology: epithelial = 0, others = 1; colon histology: no = 0, yes = 1; surgical procedure: wedge resection = 0, segmentectomy, lobectomy and pneumonectomy = 1; laser application: no = 0, yes = 1; residual disease: R0 = 0, R1/R2 = 1; the type of surgical approach: VATS = 0, open = 1; the number of resected lesions: 1 = 0, more than 1 = 1; N status: cN0pN0 = 0, cN1pN1pN2 = 1.

## Data Availability

Data supporting the findings of this study are available from the authors on reasonable request.

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
