# Peer review of "Lung Metastasectomy: Where Do We Stand? Results from an Italian Multicentric Prospective Database"

_jcm, 2024, doi:10.3390/jcm13113106_

Round 1

Reviewer 1 Report

Comments and Suggestions for Authors

Very important work with a national registry. Overall sound methods and very clear manuscript. 

Remarks: 

1) It is interesting that only few patients presented with multiple nodules. In our practice it is quite common for sarcoma patients to present with a lot of pulmonary metastases (put to 20 or 30 sometimes). What is the practice with these patients? Are they not considered for metastasectomy?

2) When you talk about the type of resection, do you count a laser enucleation as a wedge? You later mention laser but I don't see it as a factor. I think it would be worth to explore because in case of laser enucleation the surgical margin is often smaller.  "This resulted in a total of 470 surgical interventions (4 pneumonectomies, 46 lobecto-138 mies/bilobectomies, 13 segmentectomies, 407 wedge resections) and a total of 957 resected 139 metastases (median 1, range 1-36). "

3) The percentage of VATS ("With regard to surgical approach, VATS was applied in 30.4% of procedures, with no im-293 pact on post-operative morbidity") seems a bit low to me. You mentioned that 63.6% of patients had only 1 metastasis, how come these patients were operated with an open technique? Is there any consideration to this?

Author Response

Dear Reviewer

Thank you for your valuable suggestions and for your time.

We will answer to all questions point by point.

Best regards

Reviewer #1

  • It is interesting that only few patients presented with multiple nodules. In our practice it is quite common for sarcoma patients to present with a lot of pulmonary metastases (put to 20 or 30 sometimes). What is the practice with these patients? Are they not considered for metastasectomy?

Reply:

Thank you very much for your question. All patients included in this study underwent surgery with curative intent, not for diagnostic purposes. This clarification has been added to the Methods and Materials section. Only a few cases with multiple lesions were treated with a radical purpose and thus included in this study. We acknowledge the reviewer's point that in some specialized centers, multiple metastasectomies, particularly for sarcomas (sometimes involving more than 10 metastases in the lungs), may have a beneficial role in disease control, especially in pediatric patients. However, there are only a few similar cases within the cohort of patients presented in this study.

  • When you talk about the type of resection, do you count a laser enucleation as a wedge? You later mention laser but I don't see it as a factor. I think it would be worth to explore because in case of laser enucleation the surgical margin is often smaller.  "This resulted in a total of 470 surgical interventions (4 pneumonectomies, 46 lobectomies/bilobectomies, 13 segmentectomies, 407 wedge resections) and a total of 957 resected metastases (median 1, range 1-36). “

Reply:

We would like to express our gratitude to the reviewer for providing such a valuable comment and stimulating further consideration. Laser procedures, behind the number of lesions ablated, have been considered together with wedge resection, albeit this categorization may appear arbitrary due to the absence of information regarding free margin and radicality. In the majority of cases, laser ablation was employed in patients with multiple lesions (this explain the elevated number of resected metastases), although were then grouped as wedge resection. However, we recognized that laser ablation differs significantly from wedge resection (as suggested by the reviewer), so it has been separately analyzed in the multivariate assessment of factors influencing surgical and oncological outcomes. This clarification has been incorporated into the text.

  • The percentage of VATS ("With regard to surgical approach, VATS was applied in 30.4% of procedures, with no impact on post-operative morbidity") seems a bit low to me. You mentioned that 63.6% of patients had only 1 metastasis, how come these patients were operated with an open technique? Is there any consideration to this?

Reply: We extend our gratitude to the reviewer for this valuable comment. The data for this study were collected prospectively from 2010 across several Italian centers. In that period, VATS was not as widespread as it is now. Additionally, certain centers preferred the traditional approach, namely thoracotomy, to manually assess the presence of lung lesions or nodules potentially missed by preoperative imaging. (Eckardt J, Licht PB. Thoracoscopic versus open pulmonary metastasectomy: a prospective, sequentially controlled study. Chest. 2012 Dec;142(6):1598-1602. doi: 10.1378/chest.12-0249). This aspect has been highlighted into the text.

Reviewer 2 Report

Comments and Suggestions for Authors

The relevance of this rests in its capacity to provide crucial insights on the present condition and efficacy of lung metastasectomy as a therapy alternative for metastatic cancer.

Clinical Guidance: The paper is likely to provide important advice to physicians on how to make decisions about lung metastasectomy. By using a multicentric prospective database, this study may provide concrete information about patient outcomes and the variables that impact the effectiveness of the treatment in real-world scenarios. Medical practitioners may use this information to more accurately evaluate which individuals are probable to get advantages from lung metastasectomy and customise treatment regimens appropriately.
Recent Progress in Research: By including a narrative overview of current literature, the piece effectively places its results in the wider context of metastatic cancer research. This facilitates the ability of researchers and practitioners to remain informed about the most recent advancements, which include innovative methodologies, developing treatments, and changing protocols pertaining to lung metastasectomy.
Quality improvement: Articles of this kind enhance quality improvement initiatives in healthcare systems by finding opportunities for optimisation and standardisation in care delivery. The analysis of data from the prospective database and the evaluation of relevant literature may provide valuable information for the development of protocols, best practices, and quality measures for lung metastasectomy operations. This, in turn, can result in better patient outcomes and experiences.
The inclusion of an Italian multicentric database provides a global perspective to the study, allowing for a better understanding of the implementation and perception of lung metastasectomy in diverse healthcare settings. By adopting a comparative viewpoint, cooperation, information sharing, and the adoption of best practices may be facilitated across boundaries.
Patient Empowerment: It is essential for patients dealing with metastatic cancer to have a clear awareness of the latest information and available treatment choices in order to make well-informed decisions about their healthcare. Articles of this kind empower patients by offering easily understandable synopses of the most recent research discoveries, enabling them to actively participate in conversations with their healthcare professionals and engage in collaborative decision-making processes.
Overall, the paper is significant for physicians, researchers, healthcare systems, and patients because it provides valuable insights, recommendations, and evidence-based viewpoints on the use of lung metastasectomy in treating metastatic cancer.

Thank you for this article.

Comments on the Quality of English Language

Several points throughout the text need to be edited to avoid sentences that are not fully comprehensible or sentences written in not proper English. Minor English editing is necessary. 

Author Response

Reviewer #2

The relevance of this rests in its capacity to provide crucial insights on the present condition and efficacy of lung metastasectomy as a therapy alternative for metastatic cancer.

Clinical Guidance: The paper is likely to provide important advice to physicians on how to make decisions about lung metastasectomy. By using a multicentric prospective database, this study may provide concrete information about patient outcomes and the variables that impact the effectiveness of the treatment in real-world scenarios. Medical practitioners may use this information to more accurately evaluate which individuals are probable to get advantages from lung metastasectomy and customise treatment regimens appropriately.
Recent Progress in Research: By including a narrative overview of current literature, the piece effectively places its results in the wider context of metastatic cancer research. This facilitates the ability of researchers and practitioners to remain informed about the most recent advancements, which include innovative methodologies, developing treatments, and changing protocols pertaining to lung metastasectomy.
Quality improvement: Articles of this kind enhance quality improvement initiatives in healthcare systems by finding opportunities for optimisation and standardisation in care delivery. The analysis of data from the prospective database and the evaluation of relevant literature may provide valuable information for the development of protocols, best practices, and quality measures for lung metastasectomy operations. This, in turn, can result in better patient outcomes and experiences.
The inclusion of an Italian multicentric database provides a global perspective to the study, allowing for a better understanding of the implementation and perception of lung metastasectomy in diverse healthcare settings. By adopting a comparative viewpoint, cooperation, information sharing, and the adoption of best practices may be facilitated across boundaries.
Patient Empowerment: It is essential for patients dealing with metastatic cancer to have a clear awareness of the latest information and available treatment choices in order to make well-informed decisions about their healthcare. Articles of this kind empower patients by offering easily understandable synopses of the most recent research discoveries, enabling them to actively participate in conversations with their healthcare professionals and engage in collaborative decision-making processes.
Overall, the paper is significant for physicians, researchers, healthcare systems, and patients because it provides valuable insights, recommendations, and evidence-based viewpoints on the use of lung metastasectomy in treating metastatic cancer.

Thank you for this article.

Several points throughout the text need to be edited to avoid sentences that are not fully comprehensible or sentences written in not proper English. Minor English editing is necessary. 

Reply:

Thank you very much for your interesting and detailed comment. The text has been reviewed by one of the authors who is a native English speaker and has been corrected in the less comprehensible points.